# Molecular Cues for Phenological Events in the Flowering Cycle in Avocado

**DOI:** 10.3390/plants12122304

**Published:** 2023-06-13

**Authors:** Muhammad Umair Ahsan, Francois Barbier, Alice Hayward, Rosanna Powell, Helen Hofman, Siegrid Carola Parfitt, John Wilkie, Christine Anne Beveridge, Neena Mitter

**Affiliations:** 1Queensland Alliance for Agriculture and Food Innovation, The University of Queensland, Brisbane, QLD 4072, Australia; umairagri@gmail.com (M.U.A.); a.hayward@uq.edu.au (A.H.); 2School of Biological Sciences, The University of Queensland, Brisbane, QLD 4072, Australia; f.barbier@uq.edu.au (F.B.); rosanna.powell@uqconnect.edu.au (R.P.); 3Department of Agriculture and Fisheries, Queensland Government, Bundaberg, QLD 4670, Australia; helen.hofman@daf.qld.gov.au (H.H.); siegrid.parfitt@daf.qld.gov.au (S.C.P.); wilkiejd01@optusnet.com.au (J.W.)

**Keywords:** avocado, flowering genes, FT, AP1, CO, phenology

## Abstract

Reproductively mature horticultural trees undergo an annual flowering cycle that repeats each year of their reproductive life. This annual flowering cycle is critical for horticultural tree productivity. However, the molecular events underlying the regulation of flowering in tropical tree crops such as avocado are not fully understood or documented. In this study, we investigated the potential molecular cues regulating the yearly flowering cycle in avocado for two consecutive crop cycles. Homologues of flowering-related genes were identified and assessed for their expression profiles in various tissues throughout the year. Avocado homologues of known floral genes *FT*, *AP1*, *LFY*, *FUL*, *SPL9*, *CO* and *SEP2*/*AGL4* were upregulated at the typical time of floral induction for avocado trees growing in Queensland, Australia. We suggest these are potential candidate markers for floral initiation in these crops. In addition, *DAM* and *DRM1*, which are associated with endodormancy, were downregulated at the time of floral bud break. In this study, a positive correlation between *CO* activation and *FT* in avocado leaves to regulate flowering was not seen. Furthermore, the *SOC1-SPL4* model described in annual plants appears to be conserved in avocado. Lastly, no correlation of juvenility-related miRNAs miR156, miR172 with any phenological event was observed.

## 1. Introduction

Flowering is a key developmental process that evolved in angiosperms and contributes to their reproductive strategy and fitness. Plants are only able to flower once they make a switch from the juvenile to the reproductive phase. Annual plants like Arabidopsis take only a few weeks before they can flower under favourable conditions. In contrast, perennial trees face a multiple-year delay (long juvenile phase) before the onset of flowering. The tree can then cycle between vegetative growth and flowering throughout the year for several years to decades. Flowering is a demanding task and requires morphological and physiological changes in plants combined with an ability to respond to environmental factors. Various pathways have been characterized that involve the interaction of endogenous and environmental cues in regulating flowering time. Of these, the photoperiodic pathway is key, where day length is critical for flowering [1,2]. Based on their responses to photoperiod, plants can be differentiated into three groups: short-day plants (SD) (floral induction when the day length falls below the critical day length (CDL)), long-day plants (LD) (floral initiation only when day length exceeds CDL) and day-neutral plants (DN). Other pathways include age-dependent, temperature-dependent, vernalization (chill), hormonal and autonomous pathways (endogenous cues independent of external factors) [1,2,3,4,5].

*CONSTANS* (*CO*) and *Flowering Locus T* (*FT*) are key genes downstream of these pathways, including the photoperiodic, where their expression is tightly regulated by the day length [6]. *CO* encodes a zinc finger transcription factor protein and acts as a floral activator [7]. Arabidopsis flowers under long-day conditions via photoreceptors that activate *CO* transcription in the leaves [8]. *CO* then positively activates *FT* gene transcription in the leaves. *FT* is one of the most widely investigated floral genes in annual as well as perennial trees. FT belongs to a small group of proteins (involved in flowering) that are structurally similar to the mammalian phosphatidylethanolamine-binding protein (PEBPs) [9]. In Arabidopsis, this gene family comprises six members also including Terminal Flower 1 (TFL1), which shares approximately 56% amino acid identity with FT [10]. Interestingly, FT promotes flowering while TFL1 is known to repress reproductive transition [11]. Other members of the FT-related PEBP gene family include TWIN SISTER OF FT (TSF), MOTHER OF FT AND TFL1 (MFT), BROTHER OF FT AND TFL1 (BFT) and CENTRORADIALIS (CEN) [11]. AtTSF and AtMFT function redundantly with AtFT by promoting flowering [12,13], while AtBFT and AtCEN interact in flowering pathways similarly to AtTFL1 by repressing floral transition [14,15]. 

Following the activation of *FT* through CO in the leaves, the FT protein is thought to be transported to the shoot apical meristem (SAM) via the phloem [16]. In the SAM, it interacts with Flowering Locus D (bZIP transcription factor) and forms an FT–FD complex, which activates various floral meristem identity genes, including *APETALA 1* (*AP1*), *LEAFY* (*LFY*) and *FRUITFULL* (*FUL*) [16,17,18]. The MADS-box gene, *SUPPRESSOR OF OVEREXPRESSION OF CO1* (*SOC1*), is another primary flowering promoter that combines both endogenous and environmental factors in regulating flowering time and floral patterning [19,20]. *SOC1* is positively regulated by *FT*/*CO* via the FT–FD complex [21,22], gibberellin [23], and the age-dependent pathway [19]. It positively regulates the *SQUAMOSA PROMOTER BINDING PROTEIN-LIKE* (*SPL*) transcription factors (*SPL3/4/5*) to activate *AP1* and *FUL* [19,20].

Release of bud dormancy is a prerequisite to allow bud break and enable flowering to occur. Different molecular components have been identified in the control of bud dormancy in trees. The TCP transcription factor BRANCHED1 (BRC1) inhibits axillary bud outgrowth in different species and decreased *BRC1* expression is associated with bud release. In multiple perennial species, endodormancy (dormant buds to an internal plant inhibitor system) release is also strongly associated with decreased expression of *Short Vegetative Phase* (*SVP*)*/Dormancy Associated MADS-box* (*DAM*) [24,25]. Over-expression of *SVP/DAM* inhibits bud break [26,27]. Similarly, *Dormancy associated protein 1* (*DRM1*) expression decreases with dormancy release, although its exact molecular role remains elusive [28,29]. Dormancy release results in the upregulation of D-type cyclin (*CYCD*) transcripts which are involved in cell proliferation [30,31]. Regulation of bud dormancy and flowering are tightly connected and different interactions have been identified at the molecular level [26,32,33]. For example, SOC1 binds to the regulatory sequence and controls the expression of the floral repressor DAM1 in Arabidopsis, kiwi fruit and leafy spurge [34,35,36,37]. In leaf spurge, DAM1 was shown to bind to the FT promoter and inhibit its expression [36]. This interaction is not limited to perennial species. Indeed, *DAM* homologues *SVP* and *AGL24* in Arabidopsis are also involved in the repression of flowering [19]. In Arabidopsis, BRC1 was reported to directly interact with the FT protein, ref. [33] facilitating correct timing for floral bud formation by repressing premature FT-induced floral transition. 

The microRNA miR156 works antagonistically with miR172 in the age-dependent flowering pathway, where a decrease in miR156 and an increase in miR172 mark vegetative to floral phase transition during plant development [38,39,40,41]. In Arabidopsis, miR156 negatively regulates 10 out of 16 *SPL* transcripts including *SPL2/10/11*, *SPL6/13* and *SPL9/15* as well as the aforementioned flowering promoter *SPL3/4/5* [42,43]. Both *SPL3/4/5* and *SPL9* have been demonstrated to be involved in the flowering pathway [44,45,46]. *SPL3/4/5* transcripts are associated with the light-dependent flowering pathway via SOC1 and *FT* in a positive feedback loop [20,44]. The regulation of *SPL3/4/5* through *SOC1* is independent of miR156 regulation [20]. *SPL9* is known to positively regulate miR172 transcription [41] while miR172 negatively regulates six *APEALA 2*-like floral repressors, via translational inhibition [47,48,49,50]. Through the repression of *AP2*-like transcripts, miR172 promotes *FT* expression indirectly [8]. 

To date, the genetic regulation of flowering has been shown in annual as well as some perennial plants [1,51,52,53,54,55]. However, the role of these regulatory modules in regulating flowering in the vast majority of cultivated tree crops where yield depends on successful flowering requires further study. To explore how phenological changes in avocado correspond to known molecular cues, key gene transcripts were examined in the leaves and buds of avocado trees over the period of two crop cycles. Interestingly, not all floral regulators showed similar seasonal activities in avocado trees under sub-tropical Australian conditions, with respect to their role in other plant species.

## 2. Result

### 2.1. Flowering in Avocado in Queensland, Australia

Avocado is an important tree crop grown in tropical/subtropical climates around the globe. In South East Queensland (SEQ) Australia, floral bud break (visual observation) for the avocado cultivar ‘Hass’ occurs in July (mid-winter) and the trees bloom up to mid-September (spring). Vegetative growth/flush occurs twice annually—a spring flush and a summer flush(s). Following the fruit set, in October, avocado fruit development lasts until May when it is harvested commercially (Figure 1) [56,57]. The general field observations regarding inflorescence/raceme development are that avocado tends to produce flowers towards the outer canopy mostly from terminal buds.

### 2.2. Avocado Contains Multiple Copies of Floral Genes

Avocado cv Hass transcripts homologous to the PEBP (phosphatidylethanolamine-binding protein) family, including *FT*, *MFT* and *TFL1*, *CONSTANS-like* and MADS-BOX genes (*AP1/FUL*, *SEP*, *SOC1* and *SVP/DAM*), were identified from avocado transcriptomic and genomic resources available at the time of analysis (See Materials and Methods). Transcripts previously identified for *SPL* and *AP2*-like genes were also investigated [38]. Moreover, already identified *PaFT* and *PaAP1* were retrieved from NCBI [55,58]. A phylogenetic analysis was performed to gain more insight into each gene family (Figure 2, Figure 3 and Figure 4).

All the PEBP proteins were phylogenetically grouped into three clades, i.e., FT, MFT and TFL1-like (Figure 2a). Furthermore, a multiple-sequence alignment revealed structural differences among these clades (Figure 2b; the selected region is shown). For instance, the FT clade has unique amino acids tyrosine (Y), tryptophan (W) and glutamine (Q) at sites 22, 80 and 82, respectively. Similarly, the MFT clade has leucine (L), glycine (G) and proline (P) at sites 1, 33 and 51, respectively. The TFL1-like clade showed a unique amino acid sequence of glycine (G), histidine (H), isoleucine (I) and aspartic acid (D) at site numbers 13, 22, 52 and 82. These amino acids appeared conserved across different species examined in each clade.

In the phylogenetic tree constructed, AP1/FUL, SEP/AGL4, SOC1 and SVP/DAM clustered in four different clades as described previously (Figure 3a). To gain further insight into their divergence a multiple sequence alignment of their MADS-box domain was constructed using transcripts from Arabidopsis and other species alongside avocado (Figure 3b). Some amino acids were unique for one clade or two clades. For example, AP1/FUL and SEP/AGL4 have the same amino acids (glycine (G), valine (V), lysine (K) and isoleucine at sites 2, 6, 14 and 15, respectively) that were different in the other two groups. AP1/FUL has a unique amino acid isoleucine (I) at site 35. Similarly, SEP/AGL4 has unique amino acids at site 7 (glutamic acid) and site 33. The SOC1 clade has a unique amino acid, phenylalanine (F) at site 33. The SVP/DAM clade has glutamic acid, lysine, alanine, arginine, phenylalanine and threonine amino acids at sites 4, 10, 16, 26, 29 and 51, respectively, which differentiate this clade from other MADS-box clades investigated here (Figure 3b).

Two CO-like transcripts were identified in avocado from the available transcriptomic and genomic resources and used to construct a phylogenetic tree (Figure 4a) and multiple sequence alignment, revealing avocado-specific amino acids (Figure 4b). At site 1, glutamic acid (E) and glycine (G) were PaCOa and PaCOb specific, respectively.

### 2.3. Gene Expression Analysis in Avocado Leaves

Following the identification of avocado homologues of key flowering-related genes, their temporal expression patterns were examined using avocado leaves sampled from the inner and outer canopy to capture any possible impact of sunlight availability on gene expression. Heatmaps were constructed with yearly normalized Z-score values to get a glimpse of seasonal variation each year. In the leaves from the outer canopy, as shown in Figure 5a, three main clusters were distinguished. Cluster I contained *PaFT*, *PaFUL*, *PaMFTa*, *PaMFTb*, *PaSPL9a*, *PaSPL9b* and *PaRAP2.7A*. The general trends for these transcripts suggest higher abundance before the onset of visual flowering. Specifically, *PaFT* was upregulated during May/June of year 1 (2015/16). This trend was replicated for year 2 (2016/17) samples as well, almost a month before visual flowering. *PaSPL9a*, *PaSPL9b* and *PaRAP2.7A* transcript abundance patterns were similar to the *PaFT* pattern. In cluster II, *PaSPL4*, *PaSOC1a* and *PaAGL4* were grouped. The general trend for these genes was higher abundance in April/May (before visible floral induction) and December/January (fruit growth period). Cluster III contained *PaAP2*, *PaRAP2.7B*, *PaAP1*, *PaSOC1b*, *PaCOa* and *PaCOb*. This cluster further had two sub-clusters where both *CO* homologs grouped, and the remainder of the transcripts clustered together in a separate sub-group. *PaCOa* and *PaCOb* were most highly expressed just after the typical flowering period in both years (month October). This trend was more obvious during the first year. The second sub-cluster genes (*PaAP2*, *PaRAP2.7B*, *PaAP1* and *PaSOC1b*) had two expression peaks during the year 1 crop cycle, in May and September. In contrast, during the following crop year, these transcripts were most highly expressed in September only. 

Expression analysis of the selected transcripts in leaves from inside the canopy was different to that seen in the leaves from the outer canopy, suggesting a differential expression pattern between these tissues. For most of the genes, expression was high from October to December, correlating to after the flowering period each year. *FT* had a slightly similar expression trend to that observed in the outer canopy leaves (Figure 5b).

### 2.4. PaCO and PaFT Are Not Co-Regulated in the Inner and Outer Canopy Leaves

To better visualise the relationship between *CO* and *FT* transcript abundance in the outer and inner canopy leaves, the expression of both transcripts was plotted in a line graph (Figure 6). As mentioned, avocado actively flowers from July to September in SEQ, Australia. Consistent with expression analyses from avocado plants in Israel in the Northern Hemisphere, which revealed a strong transient peak in *PaFT* transcript levels in leaves of ‘Hass’ trees during early winter (end of October through November) [55], here, *FT* was also upregulated in early winter (May to June) in both years analysed. This higher transcript abundance was statistically significant compared to other time points across the year (*p* < 0.05) in which *FT* largely remained lowly expressed (Figure 6a). In addition, *PaFT* expression was higher in the outer than in the inside canopy, which correlates with flowering occurring most preferentially in the outer canopy in this species.

In contrast, *CO* had higher transcript abundance after the flowering season ended (September to October) in both years (Figure 6b). Comparing *FT* and *CO* transcript abundance, it was noted that an increase in *CO* transcript occurred almost 2 months later than the peak in *FT*. There was no apparent positive correlation between *CO* and *FT* as has been reported in Arabidopsis [59]. As observed with *PaFT*, *PaCO* expression was higher in the outer canopy.

### 2.5. SOC1-SPL4 Module

It has been previously reported that *SOC1* positively regulates *SPL3/4/5* in Arabidopsis [20]. As discussed above in Figure 5, *SOC1* and *SPL4* clustered together with similar expression patterns observed in leaves from the inside and outer canopy. To get a clearer idea, these two genes were plotted in a line graph with standard error from outer canopy leaves (Appendix A). Consistent with previous findings, their transcripts were co-expressed over the two-year crop cycle investigated here in such a way that when *SOC1* increased the transcript abundance of *SPL* also increased, and vice versa.

### 2.6. Gene Expression Analysis in Avocado Terminal Buds

Terminal buds in avocado usually turn into flowers under favourable conditions. Here an investigation of various floral identity genes including *FUL* and *AP1* with other related transcripts was conducted in terminal buds at distinct time points. Similar to the leaves, the terminal buds from the inner and outer canopy were sampled. Heatmaps generated from the yearly normalized Z-score value of the transcript abundance of each gene are shown in Figure 7. Floral identity genes (*AP1*, *LFY* and *FUL*), as well as *PaSPL9a*, *PaCYCD.3.1* and *PaAGL4*, clustered together in the illustrated heatmap due to their similar expression pattern (Figure 7a). These transcripts were highly expressed during and after the floral induction period between May and June, consistent with the previously shown trend for *PaAP1* under growing conditions in Israel [55]. This phenomenon was also replicated in the second crop cycle investigated (Year 2). This is in line with their function in floral meristem identification and bud break. No clear pattern was observed for either *SOC1* homolog. The highest expression of *PaSPL4* was observed in April each year (before floral induction). Meanwhile, transcript abundance of the dormancy-associated marker gene *DAM1* remained very low during floral induction time. *DAM1* clustered with the other dormancy-related genes *DRM1* and *BRC1*. Generally, the abundance of these dormancy-related transcripts was higher a month before actual flowering. 

A similar pattern as seen in the outer canopy buds was observed in the buds taken from inside the canopy, where *AP1*, *FUL*, *AGL4*, *CYCD.3.1* and *SPL9a* were highly expressed from July to September (during the flowering time) (Figure 7b). In contrast, dormancy-related transcripts (*DAM1*, *DRM1* and *BRC1*) were minimally abundant during flowering (July–September). Similar to results from leaves from the inside canopy, *PaSPL4* and *SOC1* in the inner canopy buds were grouped with high transcript abundance directly after flowering.

### 2.7. Abundance of miR156 and miR172 in Avocado Crop Cycle

It is known that miR156 and miR172 regulate flowering in annual plants [39,40,41]. Limited information is available on miRNA regulation during the annual crop cycle in subtropical trees. Previously, we were able to demonstrate their potential role during juvenile to adult phase transition in avocado [38]. To understand whether flowering induction and transition to flowering during the annual cycle is accompanied by fluctuation in miR156 and miR172 transcripts, these miRNAs were quantified in avocado outer canopy leaves sampled at distinctive time points before the predicted time of floral induction (early winter April–May) as well as during flowering, fruit development and fruit maturation for two consecutive years. Outer canopy leaves were sampled four times in a year i.e., spring (September), summer (December), autumn (March) and winter (May) as well as in selected samples from the phenology study described above. No significant change over different seasons in the two-year period was observed for miR156. A comparison was drawn between miR156 and putative SPL targets (*PaSPL4*, *PaSPL9a* and *PaSPL9b*), which suggested no clear pattern between miR156 and the transcript abundance of target genes (Appendix A). It is important to mention if any of the putative target genes copies are functionally correct. This, however, was not true at some time points, for example, in March 2015 when miR156/SPL expression pattern was similar. This may suggest regulation of *SPL* transcripts independent of miR156 activity as was suggested in Arabidopsis (Appendix A) [20,60]. 

In contrast, miR172 expression in the second flowering season initially decreased and was lowest in July (2015) (start of flowering), then steadily increased over time and was highest in May of the following year (harvesting time) (Appendix A). The miR172 expression results of the two consecutive years were not similar. Moreover, no clear pattern between miR172 and its targeted *AP2*-like transcripts was observed (Appendix A).

## 3. Discussion

Florigen, FT promotes floral induction in annual as well as some perennial plants and thus can be considered as a marker for floral induction [6,16,52,53,54,61,62,63]. Recently, it was suggested that *FT* is upregulated a few weeks before floral induction in trees including poplar [63], *Rhododendron × pulchrum* [62], *Fagus crenata* [54] and *Citrus* [53], and also in avocado [55]. Consistent with this, here in avocado, *FT* transcript abundance was significantly higher a few weeks before the typical timing of visible flowering in South-East Queensland Australia. It is important to note that, unlike these growing conditions, under Israeli growing conditions, avocado exhibits aggravated alternate bearing patterns, where heavy crop load in one season significantly reduces the crop yield in the following flowering season [55]. A recent finding for an avocado alternate bearing study suggests that *FT* expression in *Off* leaf (sampled from fruit-lacking trees), was significantly upregulated during the flowering induction period, as compared to On samples (leaves from fully loaded trees) [55]. These findings provide an opportunity to use *FT* transcript abundance to predict future crop yield as fluctuation in *FT* transcripts can translate into fruit yield.

FT is a key integrator in the flowering pathway and is triggered by photoperiod in Arabidopsis via *CO* [11]. Day length has an influence on CO protein activity. In long-day plants like Arabidopsis CO accumulation results in the activation of *FT*, which eventually stimulates flowering [59]. In short-day plants like rice (*Oryza sativa*) *Hd1* (*AtCO* ortholog) positively regulates *Hd3a* (*AtFT* ortholog) accumulation only in inductive conditions (short-day and long nights), while in LD conditions *Hd1* represses *Hd3a* transcripts [52]. Previous studies suggested that avocado flower induction depends on a decrease in temperature, and not necessarily on the day length [64,65]. In this study, *PaFT* mRNA accumulated in the late-autumn and early winter months in both years (temperature range of 26 °C mean maximum to 13 °C mean minimum in the years studied, Australian Bureau of Meteorology), correlating to typical floral induction time and decreased as flowering progressed. This is consistent with previous findings in ‘Hass’ avocado grown under Israeli conditions [55]. As *PaFT* expression decreased, *PaCO* mRNA accumulation started increasing and was observed highest in September (towards the end of each flowering cycle) (Figure 6). One possible hypothesis could be that *PaCO* represses *PaFT* expression under noninductive conditions (for example at the end of winter), given that CO has a role as a repressor in short-day plants under LD conditions [52]. Thus, while it must be noted that avocado floral initiation may not depend on day length, it may be possible that *CO* aids in repressing flowering at the end of each cycle (where daylength and temperature starts increasing). However, it should be noted the samples were harvested at just one time point of the day. It has been shown that there is a huge change in expression during the circadian rhythm [66]. It is possible that a window of positive correlation between *CO* and *FT* may have been missed due to the time of the day.

FT is a mobile protein, produced in the leaves and then transported into the shoot meristem where it activates floral meristem identity genes including *AP1*, *FUL* and *SEP3* [19,61,67]. *AP1* and *FUL* are closely related to MADS-box genes and are known for their key role in floral meristem identity. Phylogenetic analysis shows that avocado *AP1* and *FUL* homologs clustered in the same clade (Figure 6). In Arabidopsis, *AP1* and *FUL* are upregulated during the meristem identity transition to flowering. Here, *AP1* and *FUL* were upregulated during floral induction/bud break time (May–July, Figure 7) in avocado terminal buds. This correlated to the timing of *FT* expression upregulation in the leaves. This upregulation of *PaAP1* during the floral induction period in Queensland is consistent with the results of Ziv and co-workers (2014) in Israel [55] and with the timing of the development of floral parts investigated previously [65,68,69,70]. This could suggest that these genes are similarly required during floral initiation in these species. Further, *AGL4/SEP2*, which is shown to regulate early flower development in Arabidopsis [71], was upregulated at the time when flowering starts in avocado. These observations make them strong candidates as potential markers for floral initiation in avocado.

*SOC1* is an important floral gene involved in the genetic pathway of floral regulation [20]. In avocado, the general expression pattern of *SOC1* in buds suggested consistently highest expression in October over both years and bud locations (inner and outer), corresponding to after flowering induction (Figure 7a,b). A similar pattern was observed for *SPL4* in shaded inner avocado buds. Previously, it has been suggested that *SOC1* acts upstream of *SPL3/4/5* and regulates floral meristem identity genes including *AP1* and *FUL* through regulation of *SPL3/4/5* [20]. The clustered activity of *SOC1-SPL3/4/5* in avocado suggests that these genes may also be involved in a similar pathway to that reported in previous studies [19,20]. 

*SPL3/4/5* is a target transcript of miR156, which along with miR172 and their putative targets, are thought to regulate vegetative phase transition and flowering in annual plants [39,41,46]. This pathway not only regulates phase transition, but also plays a role in inflorescence architecture, pollen development and grain morphology [45,47,72,73]. Analysis of miR156 and miR172 profiles from avocado leaves reveal that although there was variation in miRNA transcript abundance, no conclusive evidence was found to relate this variation to any phenological event or seasonal climatic change. Comparison of miR156 expression to the targeted SPL transcripts suggests largely opposite expression patterns between them except for *SPL9a* and certain time points. It also suggests a possibility of the regulation of SPL transcript independent of miR156 regulatory function in avocado [20,60].

Out of the 10 SPL transcripts targeted by miR156 in Arabidopsis, extensive investigations focused on *SPL3/4/5* and *SPL9* regarding their role in phase transition and flowering regulation [18,39,42,74,75]. Limited information is available on the role of SPL transcripts in the yearly flowering cycle. Here, it was shown that both avocado *SPL9* homologues were generally upregulated before or during floral induction time (Figure 7). This expression pattern perfectly aligns with *SPL9* proposed role in Arabidopsis flowering where it is thought to promote *FUL* and *SOC1* transcription [76].

Finally, it should be noted that recently, *DORMANCY ASSOCIATED MADS-BOX* (*DAM*) genes are identified in various woody species and are known to play a role in bud dormancy induction [36]. *DAM* genes are closely related to *SVP* transcripts and are demonstrated to regulate *FT* to induce dormancy [36,77]. Here, in avocado, the *DAM* homolog was downregulated during the floral induction time, at the same time *FT* was upregulated. This expression pattern was similar to those of other dormancy markers (*DRM1* and *BRC1*). This suggests that *DAM* may play a role in regulating dormancy in avocado as during floral induction time bud dormancy is released to flower. In contrast to DAM, D-type cyclin *CYCD* transcripts regulate cell proliferation and are upregulated during the dormancy release [30,31]. Not surprisingly, consistent with their proposed role [30,31], the *CYCD* transcript was upregulated at the time of bud break/floral initiation in avocado. Its expression pattern was similar to the meristem identity genes (*AP1*, *FUL*) in the buds and suggest a possibility of coregulation among these transcripts.

## 4. Materials and Methods

### 4.1. Transcript Identification

The NCBI database was thoroughly searched to find already characterized gene transcripts. *PaAP1* and *PaFT* sequences were retrieved from the NCBI database [55]. Moreover, *SPL*-like, *AP2*-like and *SOC1* transcripts and primers from Ahsan et al. [38] were used. Arabidopsis homologs were used to make BLAST searches of the available transcriptomic data [78], leading to the potential candidate transcripts using Geneious ver. 11. These potential transcripts were then translated into all possible combinations using Geneious’s in-built translator. The correct frame was identified by aligning the translated transcripts to Arabidopsis proteins. Pfam online tool was employed to confirm the integrity of the protein domain in the identified transcripts. To get more confidence, reciprocal BLAST on TAIR Arabidopsis and NCBI online databases were completed. Further, Open Reading Frame (ORF) was predicted by comparing and aligning similar transcripts from all available transcriptomic databases and was verified using the NCBI ORF finder online. These newly identified transcripts were deposited in the NCBI public database (Appendix A).

### 4.2. Phylogenetic Analysis

To illustrate the divergence of *FT*-related PEBP, *CO*-like and MADS-box (*AP1*, *SOC1*, *SEP* and *SVP/DAM*) genes, phylogenetic trees were constructed in MEGA 7 following Maximum Likelihood phylogenetic analysis parameters with defaults setting. To analyse structural variations of amino acids in proteins, multiple sequence alignment was generated in Geneious ver. 11, using the muscle alignment default parameter.

### 4.3. Tissue Collection, Handing, Grinding

Four avocado trees of cv. Hass grafted on cv. Velvick grafted on cv. H2 were sampled from the Bundaberg area of Queensland, Australia. They were sampled every 4 to 6 weeks for 2 consecutive years (for miRNA year 1 samples were collected every 3 months). The age of the trees was approximately 10 years and they were maintained under industrial orchard conditions. Leaves from the inside and outer canopy were collected. As avocado flowers on terminal buds, terminal buds from the inside and outer canopy were collected.

The general sampling strategy was to collect leaves and bud tissues from 15 different positions on the trees to minimize variation. The samples were collected between 9 a.m. and 11 a.m., and the sampling time remained consistent throughout the study. All tissue samples were placed on dry ice quickly after detaching them from the trees and then they were subsequently transported to the lab where they were immediately placed in a −80 °C freezer. The tissues were then cryogenically ground to a fine powder using a Geno grinder and were dispensed to 2 mL tubes for downstream analysis (SPEX SamplePrep, 2010 Geno/grinder, (Metuchen, NJ, USA).

### 4.4. RNA Extraction, cDNA Synthesis and qRT-PCR for miRNA Analysis

The total RNA from the outer canopy leaves was extracted using the MasterPure Plant RNA purification kit by manufacturer instruction (Epicentre, Madison, WI, USA). The integrity of the RNA was tested using 1% TAE agarose gel and was then quantified using a Nanodrop ND-1000 spectrophotometer (Thermo scientific, Waltham, MA, USA). A total of 250–500 ng of high-quality RNA was used to prepare low molecular weight cDNA synthesis using a miSCRIPT Plant RT kit per manufacturer protocol (Qiagen, Venlo, The Netherlands). This prepared cDNA was then used to run qRT-PCR reactions in technical duplicates using Quantitect SYBR green kit according to manufacturer instructions (Qiagen, The Netherlands). The run was performed on a Roter-Gene Q-6000 machine (Qiagen, The Netherlands). U6-snoRNA and 5.8S rRNA were used to normalize miR156 and miR172 transcript abundance. Primers were used to amplify transcripts from Ahsan et al. [79].

### 4.5. RNA Extraction, cDNA Synthesis and qRT-PCR for Gene Quantification

For gene quantification, total RNA was extracted using a phenol/chloroform-free modified CTAB-based nucleic acid extraction method [80]. A Bio-Rad iScript cDNA synthesis kit was used to prepare high molecular weight cDNA. The transcript abundance was quantified by qRT-PCR in CFX384 thermal cycler (Bio-Rad, Hercules, CA, USA) using a SensiFAST™ SYBR^®^ No-ROX Kit (Bioline, London, UK) as per manufacturer instructions. Avocado orthologs of commonly used housekeeping genes (*GAPDH*, *EF1a*, and *PP2AA3*) were utilized to calculate relative expression [38,81]. Primers used to amplify transcripts are shown in Appendix A and Ahsan et al. [38] (Appendix A).

### 4.6. Data Analysis

Relative abundance was calculated using the delta-delta-Ct method with corrected primer efficiency calculated through Linreg PCR ver. 7.5 (the University of Amsterdam, The Netherlands) (Relative abundance = Gene PE^ (−Gene Ct)/Control PE^ (−Control Ct). One-way ANOVA (analysis of variance) with Tukey test was done to compute statistical significance difference. GraphPad Prism 6 was used to plot the relative abundance of each transcript. Further, *Z*-score was calculated using the formula, z = x − μ/σ, where, x is the relative abundance of a time-point, μ is the mean of all samples, and σ is the standard deviation. To see the seasonal variation in a crop cycle, Z-score was normalized only with samples from that year. To get a year-to-year abundance pattern overall Z-score (of two years) was calculated. The heatmaps were visualized using the heatmapper.ca online tool [82].

## 5. Conclusions

In conclusion, here we identified key floral regulatory genes including *FT*, *CO*, *FUL*, *SOC1*, *DAM* and *AGL4* from avocado. Phylogenetic analysis showed that these identified genes share the conserved protein domain with their Arabidopsis counterpart. Based on the expression data, these identified transcripts were correlated with important phenological events or seasonal climatic changes in the annual crop cycle. Based on this, here we suggest a few transcripts including *FT* (similar to previously discussed *PaFT* function as a florigen [55]), *FUL* and *AP1* that could potentially be used as potential markers for floral induction in avocado. These outcomes may create an opportunity to discuss these molecular factors and their role in tree flowering. This genetic information may be helpful to understand bottlenecks in flowering, breeding and ultimately productivity of these horticultural trees, where crop cycle and flowering heavily influence yield.

## Figures and Tables

**Figure 1 plants-12-02304-f001:**
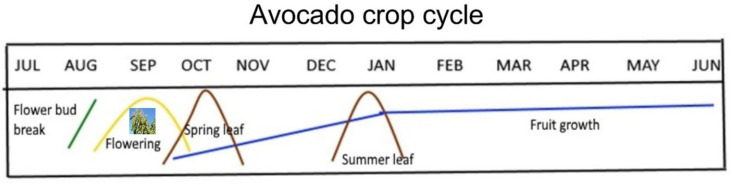
Avocado ‘Hass’ crop phenology cycle under South East Queensland (Australia) conditions. The illustrations were created using the actual average phenotypic data for 2 years (2015–2017) with modifications from the avocado crop cycle [57].

**Figure 2 plants-12-02304-f002:**
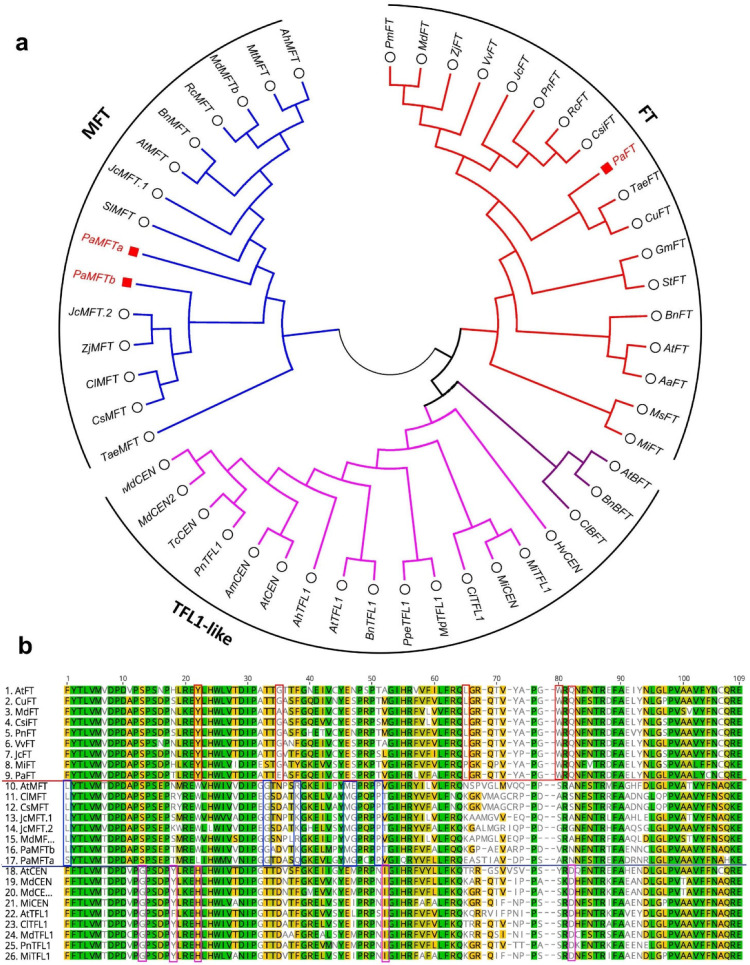
Phylogenetic analysis of FT-related PEBP family genes. (**a**) PEBP family proteins phylogenetic tree constructed with Maximum likelihood analysis of already published sequences from other crops (Appendix A) with identified FT-related transcripts from avocado. (**b**) Sequence alignment of the PEBP domain of the FT, MFT and TFL1-like. The unique or dissimilar sequence from each clade is shown in the box.

**Figure 3 plants-12-02304-f003:**
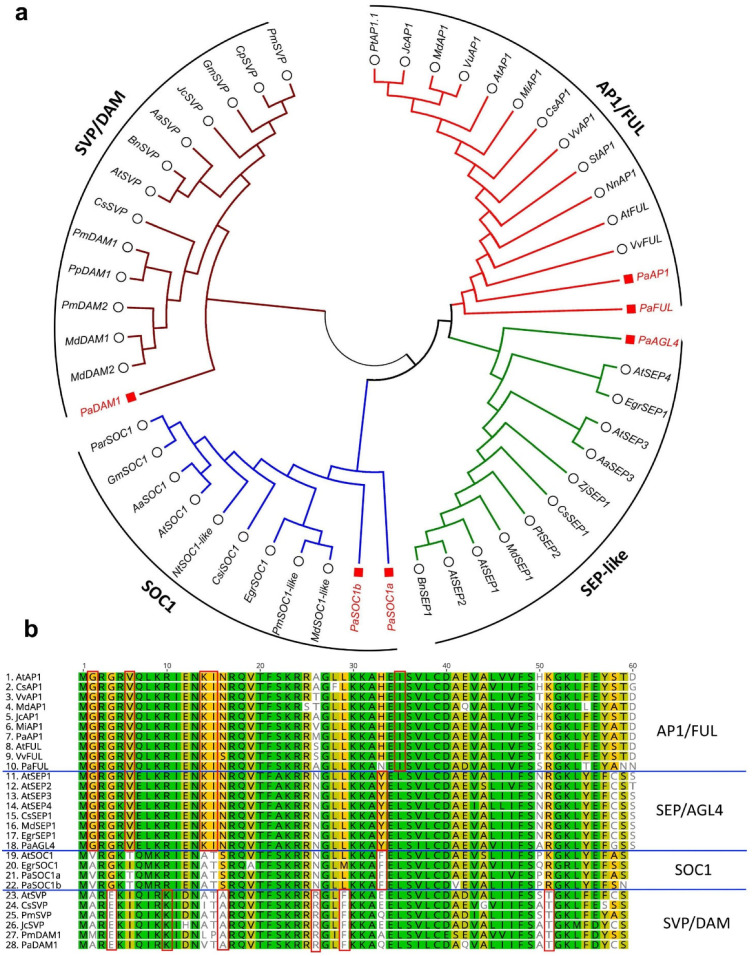
Phylogenetic analysis of MADS-box protein family 4 clades. (**a**) phylogenetic tree constructed with Maximum likelihood analysis of already published sequence from other crops (Appendix A) with identified MADS-box family transcripts from avocado. (**b**) Sequence alignment of the MADS-box domain of AP1/FUL, SEP, SOC1 and SVP/DAM. The unique sequence from each clade is shown in the box.

**Figure 4 plants-12-02304-f004:**
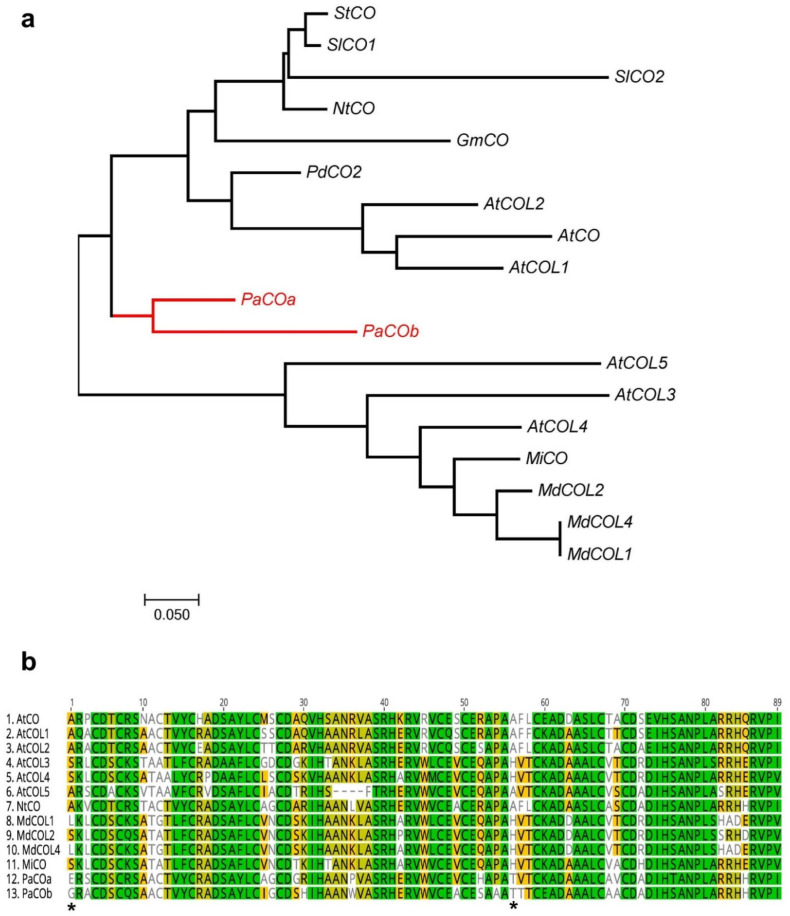
Phylogenetic analysis of CONSTANS-like genes from different plants including avocado. (**a**) phylogenetic tree constructed with Maximum likelihood analysis (See Materials and Methods section for tree parameters), (**b**) Sequence alignment of highly conserved B-box zinc finger domain region of CO-like transcripts with avocado’s identified transcripts (Appendix A). Symbol (*) is used to illustrate avocado-specific amino acids.

**Figure 5 plants-12-02304-f005:**
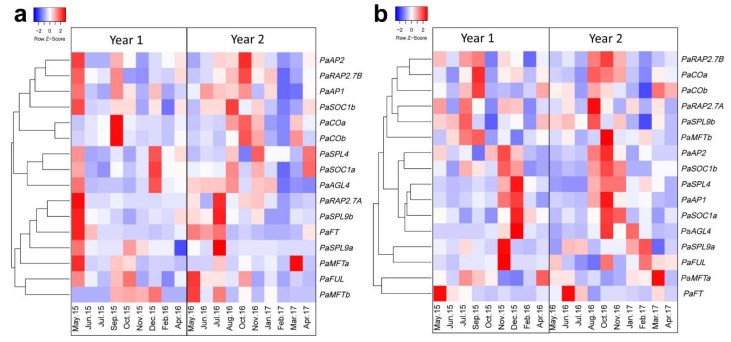
Yearly gene expression pattern in avocado leaves. Heatmaps of the relative abundance of transcripts in the leaves from (**a**) outer canopy and (**b**) inside canopy. Yearly normalized *Z*-score was used to construct heatmaps in a colour scale (where Red, represents higher expression and Blue, low abundance) to show transcript variation in a crop cycle (See Materials and Methods for heatmaps parameters).

**Figure 6 plants-12-02304-f006:**
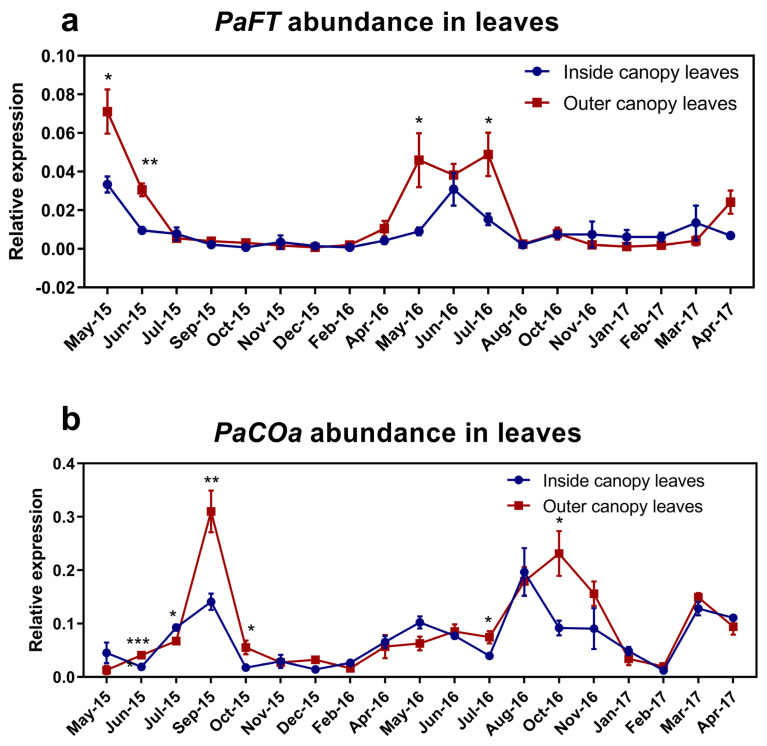
Transcript abundance comparison between inner and outer canopy leaves in avocado. The analogy of (**a**) PaFT and (**b**) PaCOa was drawn to check relative transcript abundance between the inside and outside canopy. Error bars represent the standard error of the mean (*n* = 4), and significant differences calculated by one-way ANOVA are shown by an asterisk (*p* < 0.05 = *, *p* < 0.01 = **, *p* < 0.0001 = ***).

**Figure 7 plants-12-02304-f007:**
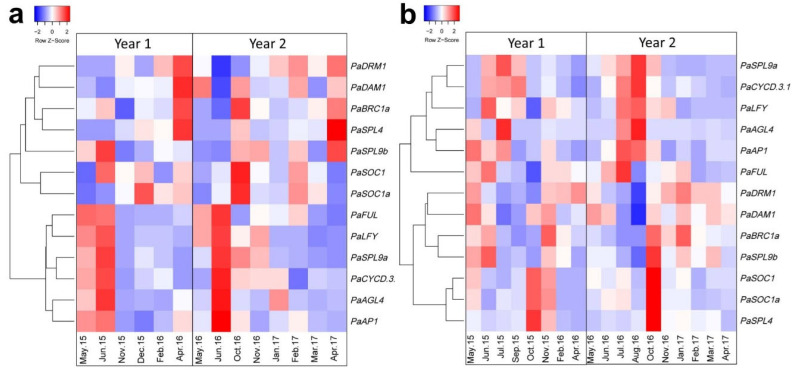
Yearly gene expression pattern in avocado terminal buds. Heatmaps of the relative abundance of transcripts in the terminal buds from (**a**) outer canopy (facing sun) and (**b**) inside canopy (shade). Yearly normalized *Z*-score was calculated from qRT-PCR data and was used to construct heatmaps in a colour scale (Red = higher expression, Blue = low expression) to show transcript variation in a crop cycle (See Materials and Methods for heatmaps parameters).

## Data Availability

Not applicable.

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
