# Peer review of "Molecular Cues for Phenological Events in the Flowering Cycle in Avocado"

_plants, 2023, doi:10.3390/plants12122304_

Round 1

Reviewer 1 Report

This study examines the molecular cues that potentially regulate the yearly flowering cycle in avocado for two consecutive crop cycles. The researchers identified homologues of flowering-related genes in avocado and assessed their expression profiles in various tissues throughout the year. They suggest that the identified genes, including FT, AP1, LFY, FUL, SPL9, CO, and SEP2/AGL4, are potential candidate markers for floral initiation in avocado crops. The study also found that the DAM homolog was downregulated during the floral induction time, simultaneously as FT was upregulated, implying that DAM may play a role in regulating dormancy in avocado. These findings contribute to our understanding of the molecular events underlying the regulation of flowering in tropical tree crops and may help enhance horticultural tree productivity.

The study offers valuable insights into the genetic regulation of flowering in avocados, potentially impacting the agricultural industry. The identification of key floral regulatory genes and their conserved protein domains shared with Arabidopsis counterparts significantly contributes to plant genetics knowledge.

However, the paper could benefit from a more detailed explanation of the methodology, particularly concerning sample collection and analysis. The authors mention that samples were collected at a single time point of the day, potentially missing significant changes in gene expression during the circadian rhythm. It would be beneficial to specify the time of day when samples were taken and whether it was consistent throughout the study.

Additionally, the authors need to indicate which reference gene was used to normalize the qPCR data and justify their choice.

In conclusion, this paper presents a valuable contribution to plant genetics research. However, further investigation is required to fully comprehend the intricate genetic mechanisms that regulate flowering in avocados.

Author Response

Dear Reviewer,

Thanks for your valuable feedback. We have made changes as recommended by you and are incorporated in the manuscript.

The following has been incorporated:
In methods section line 401-402:
"The samples were collected between 9 a.m. and 11 a.m., and the sampling time remained consistent throughout the study."

The second recommended incorporation was for reference genes under Method section at line 425-427 and we have incorporated the details of the house keeping genes, as follow:
“Avocado orthologs of commonly used housekeeping genes (GAPDH, EF1a, and PP2AA3) were utilized to calculate relative expression. (reference updated in the manuscript).”

Reviewer 2 Report

Dear Authors 

I have read with great interest the experimental data presented in the manuscript entitled "Molecular cues for phenological events in the flowering cycle in avocado." The work aims to identify and study the expression intensity of gene homologues (FT, CO, FUL, SOC1, DAM and AGL4) involved in flowering regulation in tropical trees using avocado as an example. In contrast to annual plants, the molecular mechanisms of flowering in woody plants are poorly understood. This makes the work relevant. The authors used adequate research methods. The manuscript contains new data. Comparison of flowering gene transcript profiles with phenological events in the annual cycle revealed at least three gene homologues, including FT, FUL and AP1, which are proposed to be used as possible markers of flowering induction. It is quite logical that the activity of DAM and DRM1 gene homologues was suppressed during flower bud blossoming. Interestingly, in avocado plants there was no positive correlation between the levels of CO and FT gene transcripts in leaves during the flowering cycle, nor was any correlation found between the levels of juvenility-related miR156 and miR172 with the phenological events of the flowering process.

 In the text of the manuscript and in the appendix, I could not find the names of the reference genes that were used in the work to normalize the transcript levels of the flowering genes studied.

 The manuscript can be accepted as presented.

Author Response

Dear Reviewer,

Thanks for your valuable feedback. I have included reference genes (Methods section line 425-427) that we used in this study and added the primer sequence of reference genes in the appendix. 

Thanks again.